# Quorum sensing via dynamic cytokine signaling comprehensively explains divergent patterns of effector choice among helper T cells

**Edward C. Schrom II**[ID]*, **Simon A. Levin**, **Andrea L. Graham**

Department of Ecology and Evolutionary Biology, Princeton University, Princeton, New Jersey, United States of America

* eschrom@princeton.edu

**Data Availability Statement:** All relevant data are within the paper and its Supporting Information files. All results can be reproduced from the equations provided in the text and supporting

## Abstract

In the animal kingdom, various forms of swarming enable groups of autonomous individuals to transform uncertain information into unified decisions which are probabilistically beneficial. Crossing scales from individual to group decisions requires dynamically accumulating signals among individuals. In striking parallel, the mammalian immune system is also a group of decentralized autonomous units (i.e. cells) which collectively navigate uncertainty with the help of dynamically accumulating signals (i.e. cytokines). Therefore, we apply techniques of understanding swarm behavior to a decision-making problem in the mammalian immune system, namely effector choice among CD4+ T helper (Th) cells. We find that incorporating dynamic cytokine signaling into a simple model of Th differentiation comprehensively explains divergent observations of this process. The plasticity and heterogeneity of individual Th cells, the tunable mixtures of effector types that can be generated *in vitro*, and the polarized yet updateable group effector commitment often observed *in vivo* are all explained by the same set of underlying molecular rules. These rules reveal that Th cells harness dynamic cytokine signaling to implement a system of quorum sensing. Quorum sensing, in turn, may confer adaptive advantages on the mammalian immune system, especially during coinfection and during coevolution with manipulative parasites. This highlights a new way of understanding the mammalian immune system as a cellular swarm, and it underscores the power of collectives throughout nature.

## Author summary

Across the animal kingdom, swarming is a common phenomenon by which many autonomous individuals act as a unified group. Similarly, helper T cells in the mammalian immune system are numerous and autonomous, and yet they collectively make important decisions, such as which immune weapons to recruit during a given infection (i.e. "effector choice"). However, due to varying experimental results, it is unclear when, how, and why helper T cells coordinate unified effector choices. Inspired by studies of swarms in

information. Code the authors used for obtaining all results and figures are publicly available at https://github.com/eschrom/T-Cell-Quorum-Sensing.

**Funding:** ECS was funded by the National Science Foundation Graduate Research Fellowship Program (DGE-1656466) (https://www.nsfgrfp.org/) and by the Princeton Center for Health and Wellbeing (https://chw.princeton.edu/). The funders had no role in study design, data collection and analysis, decision to publish, or preparation of the manuscript.

the animal kingdom, we answer all three questions with a single set of simple mathematical rules governing the interactions of individual cells. Helper T cells engage in quorum sensing, transitioning from mixed to unified group decisions only at high cell densities. Quorum sensing emerges naturally from the interplay between molecular circuits within helper T cells and dynamically accumulating signals between helper T cells. Quorum sensing may have evolved because it helps our immune systems discern legitimate changes in effector needs from parasitic sabotage of the effector choice system. These insights demonstrate that the quantitative study of swarm biology can shed new light on the organization and function of the mammalian immune system.

## Introduction

Collective behavior–the coordinated action of many autonomous individuals–can accomplish sophisticated information-processing tasks that may be impossible for lone individuals. This has led to the repeated evolution of swarming across various taxa [1]. For example, honeybee swarms leverage multiple types of interactions among individuals to choose the best nesting site among several options [2,3]. Ant swarms leverage variability in chemical signaling among individuals to dynamically track moving food sources [4,5]. Bacterial swarms use quorum sensing–a special class of collective behavior in which different group decisions emerge depending on the density of constituent individuals–to measure and respond in unison to fluctuating environmental conditions [6,7]. In each example, collective behavior allows swarms to integrate conflicting, changing and otherwise uncertain information into unified decisions which are dynamically updated and probabilistically beneficial.

Although swarms are typically considered to comprise distinct organisms, collective behavior can also arise from cells within an organism. In particular, the mammalian immune system embodies many aspects of collective behavior. Immune cells are decentralized and autonomous individuals that together make coherent decisions despite substantial uncertainty [8,9]. For example, CD4+ helper T (Th) cells collectively decide whether a foreign invader warrants an immune response, and which immune effectors to deploy. Understanding how such decisions emerge requires understanding how cells collectively coordinate their behavior [10,11]. Just as in insect swarms, this communication involves complex feedbacks within and among Th cells [12,13], which vary drastically in their signaling outputs [14,15,16,17]. Thus, we propose that the lens of collective behavior may reveal novel insights into how the immune system processes uncertain, conflicting, and changing information [8,9].

We apply that lens here to study Th effector choice. This process begins when sentinels called antigen-presenting cells (APCs) enter lymph nodes, bearing fragments of parasites called antigens. Th cells that recognize and bind these antigens form immunological synapses with APCs, through which they receive instructions to proliferate and differentiate into a given effector type (e.g. Th1, Th2, Th17) [18]. These types correspond to different classes of infection; for example, Th1 cells combat intracellular microparasites, while Th2 cells combat extracellular macroparasites [18]. Each Th cell broadcasts its type to its neighbors via diffusible signaling molecules called cytokines, influencing their effector differentiation [19,20]. At the Th group scale (e.g., across a lymph node), accurate differentiation into the effector type best matched to the current threat is critical for host survival [21,22,23].

Effector choice is difficult for several reasons. First, information is limited: APCs are rare, such that each Th cell has a low probability of receiving effector instruction from an APC on a per-hour basis [24]. After APC contact, a Th cell resists further contact for up to 72 hours,

precluding ongoing APC instruction [25]. Second, information may be conflicting: because mammalian hosts in nature are constantly coinfected with parasites requiring different effector responses [26], APCs that have encountered different types of parasites will instruct for different effector types. Third, information may be changing and deceptive: many parasites manipulate APCs to instruct for inaccurate effector types in order to escape clearance [27,28,29]. At first glance, cytokine signaling among Th cells may only amplify this uncertainty. It is unclear how Th cells process conflicting and potentially untrustworthy information.

Furthermore, Th cells seem to process information differently in different settings. *In vivo*, Th cells often make strictly polarized decisions. Coinfections with parasites requiring different effector responses often elicit unified commitment to one effector type exclusively [30,31,32]. On the other hand, *in vitro*, Th cells given conflicting effector stimulation adopt mixed effector types, simultaneously secreting cytokines characteristic of different effector types [17,33,34]. These contradicting results are difficult to reconcile. Given that Th cells can plastically switch effector types [13,35,36,37] and display broad cell-to-cell variability in their cytokine expression [14,15,16,17], the strictly polarized decisions that arise despite conflicting APC instruction *in vivo* seem especially difficult to explain.

Here, we solve this immunological puzzle by understanding Th cells as a swarm. Through collective behavior, group consensus and commitment can arise despite conflicting environmental cues via dynamic signals that cross scales from the individual to the group. Whether these signals are autoinducers secreted by bacteria [6], pheromones deposited by ants [4], or even startle responses among schooling fish [38], what matters is that they dynamically accumulate. By analogy, Th cells are individuals, all Th cells in a lymph node form a group, APCs are (possibly) conflicting environmental signals, and cytokines are the dynamically accumulating signals among individuals. With this motivation, we modified a well-studied model of the gene expression motif driving Th1 vs. Th2 differentiation [33,39,40,41] to include the key cytokines in this process. We then addressed four Questions:

1. Does the model explain how mixed Th effector types arise *in vitro*?

2. Does the same model explain how polarized group effector decisions arise *in vivo*?

3. When does dynamic cytokine signaling matter *in vivo*, given the presence of APCs?

4. What advantages could collective coordination via dynamic cytokine signaling provide?

We find that polarized group effector decisions emerge only above a threshold cell density, simultaneously explaining *in vitro* and *in vivo* observations, and epitomizing quorum sensing. Moreover, our model predicts that quorum sensing operates even in the presence of APCs and leverages cell-to-cell variability in cytokine signaling to discern true from deceptive information. Indeed, it has recently been suggested that Th cells use quorum sensing to make other immune decisions [42,43], and one group has provided empirical evidence that Th cell density modulates the rate of memory differentiation [44]. Empirical studies have also demonstrated that quorum sensing regulates key processes in other closely related immune cells, such as CD8+ killer T cell proliferation [45] and B cell motility [46]. Here, we provide the first comprehensive explanation of how a quorum emerges from a group of Th cells, and why Th quorum sensing might adaptively benefit the host organism.

## Model development

Although Th effector differentiation is the result of a complex gene expression program, the simplifying assumption that the transcription factor T-bet primarily drives Th1 differentiation and the transcription factor GATA3 primarily drives Th2 differentiation is both common and

empirically supported [33,39,40,41]. Both transcription factors induce their own expression ("self-activation") and diminish each other's expression ("cross-inhibition"), forming a previously analyzed transcription factor motif [33,39,40,41]. While these transcription factors are confined within Th cells, Th1 and Th2 cells also secrete cytokines–IFNγ and IL-4, respectively–that diffuse through the extracellular space and similarly self-activate and cross-inhibit [13]. We searched the immunological literature for all known molecular interactions among these transcription factors and cytokines. Together, they form a system of four ordinary differential equations ("ODEs") (Fig 1 and S1 Table, see S1 Text regarding mathematical forms). The parameter values assigned to each interaction were also grounded in the immunological literature (S1 Table); nonetheless, sensitivity analyses show that errors in these parameter estimates do not qualitatively alter our results (S2 Text). Thus, these ODEs and parameter values describing average molecular expression across a group of Th cells are the foundation of our

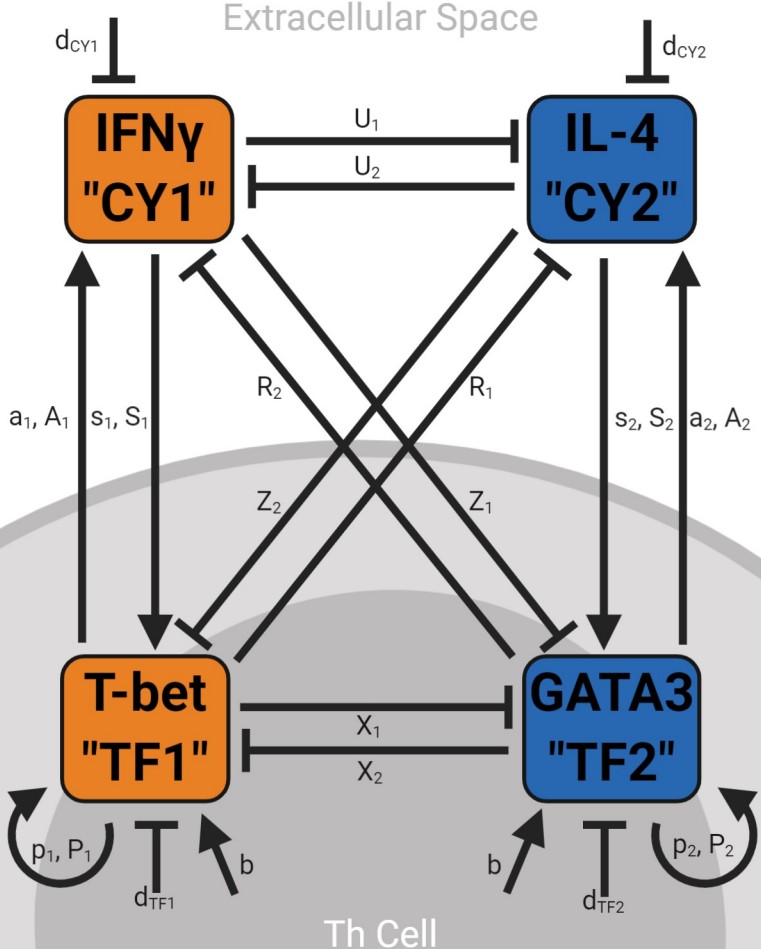

**Fig 1. Model schematic.** T-bet and GATA3 are the master transcription factors controlling Th1 and Th2 differentiation, respectively, and are confined within Th cells. IFNγ and IL-4 are the master cytokines controlling Th1 and Th2 differentiation, respectively, and are free to diffuse through the extracellular space. Together, expression of these four molecules are the four state variables of the dynamic model. Each of these four molecules can upregulate (arrow-head interactions) or downregulate (T-head interactions) the expression of the other molecules in the model. References to the immunological literature supporting the existence of the depicted interactions and their assigned parameter values are provided in S1 Table.

analyses.

$$\mathrm{d}\boldsymbol{TF}_1 = \left[\left(b + \frac{p_1\boldsymbol{TF}_1^{h_p}}{P_1^{h_p} + \boldsymbol{TF}_1^{h_p}}\right)\left(\frac{X_2^{h_x}}{X_2^{h_x} + \boldsymbol{TF}_2^{h_x}}\right) + \left(\frac{s_1\boldsymbol{CY}_1^{h_s}}{S_1^{h_s} + \boldsymbol{CY}_1^{h_s}}\right)\left(\frac{Z_2^{h_z}}{Z_2^{h_z} + \boldsymbol{CY}_2^{h_z}}\right) - d_{\mathrm{TF1}}\boldsymbol{TF}_1\right]dt \quad \text{Eq 1}$$

$$\mathrm{d}\boldsymbol{TF}_2 = \left[\left(b + \frac{p_2\boldsymbol{TF}_2^{h_p}}{P_2^{h_p} + \boldsymbol{TF}_2^{h_p}}\right)\left(\frac{X_1^{h_x}}{X_1^{h_x} + \boldsymbol{TF}_1^{h_x}}\right) + \left(\frac{s_2\boldsymbol{CY}_2^{h_s}}{S_2^{h_s} + \boldsymbol{CY}_2^{h_s}}\right)\left(\frac{Z_1^{h_z}}{Z_1^{h_z} + \boldsymbol{CY}_1^{h_z}}\right) - d_{\mathrm{TF2}}\boldsymbol{TF}_2\right]dt \quad \text{Eq 2}$$

$$\mathrm{d}\boldsymbol{CY}_1 = \left[\left(\frac{a_1\boldsymbol{TF}_1^{h_a}}{A_1^{h_a} + \boldsymbol{TF}_1^{h_a}}\right)\left(\frac{R_2^{h_r}}{R_2^{h_r} + \boldsymbol{TF}_2^{h_r}}\right)\left(\frac{U_2^{h_u}}{U_2^{h_u} + \boldsymbol{CY}_2^{h_u}}\right) - d_{\mathrm{CY1}}\boldsymbol{CY}_1\right]dt \qquad \text{Eq 3}$$

$$\mathrm{d}\boldsymbol{CY}_2 = \left[\left(\frac{a_2\boldsymbol{TF}_2^{h_a}}{A_2^{h_a} + \boldsymbol{TF}_2^{h_a}}\right)\left(\frac{R_1^{h_r}}{R_1^{h_r} + \boldsymbol{TF}_1^{h_r}}\right)\left(\frac{U_1^{h_u}}{U_1^{h_u} + \boldsymbol{CY}_1^{h_u}}\right) - d_{\mathrm{CY2}}\boldsymbol{CY}_2\right]dt \qquad \text{Eq 4}$$

We made several modifications to this basic ODE model in order to address the four Questions outlined above. First, Question 1 requires studying not just the mean but the full distribution of molecular expression across a group of Th cells. Thus, we extended our model into a system of four stochastic differential equations ("SDEs"), by adding to each equation the differentials of independent Brownian motion processes ($+[n_{TF}\boldsymbol{TF}_1]dW_{TF1}$ for Eq 1, and analogously for Eqs 2–4, see S3 Text). Because mammalian cells express proteins in nearly discrete bursts [47,48] such that any given cell fluctuates across the entire distribution of expression through time [49,50], SDEs are an appropriate mathematical tool [51]. In fact, the specific form of our stochastic term appropriately models lognormal fluctuations in molecular expression, because distributions of T-bet, GATA3, IFNγ, and IL-4 expression among Th cells span several orders of magnitude with large positive skew [34,52]. Therefore, many simulated sample paths of these SDEs together approximate the distribution of molecular expression in a group of Th cells.

Second, Question 2 requires comparing Th cells *in vitro* vs. *in vivo*. A major difference between these settings is cell density: *in vitro* Th culture requires ~$10^6$ cells/mL [33,34], whereas Th cells exist *in vivo* in lymph nodes at ~$10^9$ cells/mL [20]. Thus, we extended our model to accommodate this range of cell densities, by identifying which parameters depend on cell density. While intracellular transcription factors are measured as the number of copies per cell and therefore do not depend on cell density, extracellular cytokines are measured in terms of concentration in the extracellular space and do depend on cell density. As cell density increases, the proportion of extracellular space decreases, compacting secreted cytokines into smaller volumes. Thus, in terms of extracellular concentration, both cytokine production ($a_{1,2}$) and removal ($d_{CY1,2}$) rates increase with increasing cell density (S1 Fig and S1 Text). Moreover, production is driven only by cellular secretion, but removal is driven by both cellular consumption and free decay (which is often fast for molecules involved in cytokine regulation [53]). Therefore, over the range of cell densities we studied, $a_{1,2}$ scales more steeply with cell density than does $d_{CY1,2}$. Consequently, cytokines dynamically turn over faster and accumulate to higher levels as cell density increases (S1 Fig). See S1 Text for a full explanation of the units and cell density dependencies in this model.

Third, both Questions 3 and 4 require accounting for instruction by APCs. Biologically, APCs provide Th1 or Th2 instruction by secreting Th1- or Th2-driving cytokines directly onto a Th cell surface via an immunological synapse [18,54,55,56]. Therefore, we included

APC instruction by augmenting every appearance of $CY_1$ and $CY_2$ (except decay) in the model equations with constants $APC_1$ and $APC_2$, whose values depend on the frequencies of Type 1 and 2 APCs, respectively (see S1 Table). For example, $\left(\frac{s_1 CY_1{}^{h_s}}{s_1{}^{h_s}+CY_1{}^{h_s}}\right)$ becomes $\left(\frac{s_1 (CY_1+APC_1)^{h_s}}{s_1{}^{h_s}+(CY_1+APC_1)^{h_s}}\right)$ in Eq 1, and so forth. This allows APCs to influence Th decision-making without following the same rules of production and removal as cytokines.

## Results

In answering the four Questions above, we find that our model explains the divergent behavior of Th cells in different settings. The flexibility and heterogeneity of individual Th cells observed *in vitro* as well as the polarization of Th groups observed *in vivo* emerge from the same set of underlying molecular rules. Among these rules, dynamic cytokine signaling is the key to all behaviors. Because the role of dynamic cytokine signaling in group behavior depends on cell density, Th effector choice is a clear example of quorum sensing.

### Dynamic cytokine signaling drives mixed Th effector types *in vitro*

*In vitro*, Th cells differentiate into highly heterogeneous mixtures of Th1, Th2, and Th1-Th2 hybrid effector types [33,34]. In fact, at the individual cellular scale, there is a nearly uniform distribution of cells that are fully Th1 (only express T-bet), fully Th2 (only express GATA3), or some mixture of the two (express both T-bet and GATA3 at varying levels) (Fig 2A inset, from [34]). Using 1000 sample paths of our SDE model to predict the distribution of Th1 vs. Th2 transcription factor expression across cells under these conditions, we also observe a nearly uniform distribution (Fig 2A). In contrast, when the same experiment is run with αIFNγ and αIL-4 antibodies to block cytokine signaling, this uniform distribution transforms into a U-shaped distribution, in which Th cells express either T-bet or GATA3, but few express both (Fig 2B inset, from [34]). When cytokine secretion is eliminated, our SDE model also correctly predicts this U-shaped distribution (Fig 2B).

These agreements between our stochastic model and experimental data can be explained by the underlying equilibrium behavior of our deterministic model. Under normal model parametrization at $10^6$ cells/mL, cytokine accumulation drives stable coexpression of T-bet and GATA3. However, as the degree of permitted cytokine secretion is reduced, as by cytokine-blocking antibodies, model terms involving cytokines fade in importance compared to terms involving only transcription factors. This drives a bifurcation, in which transcription factor expression becomes bistable–only pure Th1 or pure Th2 cell types are stable cellular states (Fig 2C). Thus, our model's equilibrium behavior explains the experimental finding that some degree of dynamic cytokine signaling is required to produce mixed Th effector types.

In addition to equilibrium behavior, our model also reproduces the dynamics by which mixed Th effector types arise *in vitro*. Mean cellular levels of T-bet and GATA3 grow non-linearly through time and eventually favor Th2, a pattern which is captured well by our ODE and SDE models (Fig 3). In fact, the timescale of these dynamics resolves a conflict between experiment and prior theory. Empirically, varying inputs of exogenous Th1- and Th2-stimulation to 7-day Th cultures leads to a continuum of T-bet and GATA3 expression levels which appears stable [17,33]. But mathematically, merely varying input conditions to a dynamical system cannot change where its stable equilibria lie. Our model agrees with experimental data in that varying inputs of IFNγ and IL-4 create a continuum of T-bet and GATA3 expression after 7 days (S2A and S2B Fig). But our model also shows that although these tunable states appear stable, they are truly transient and eventually converge on a common mixed effector type (S2C–S2F Fig). This common mixed effector type is not observed empirically because it

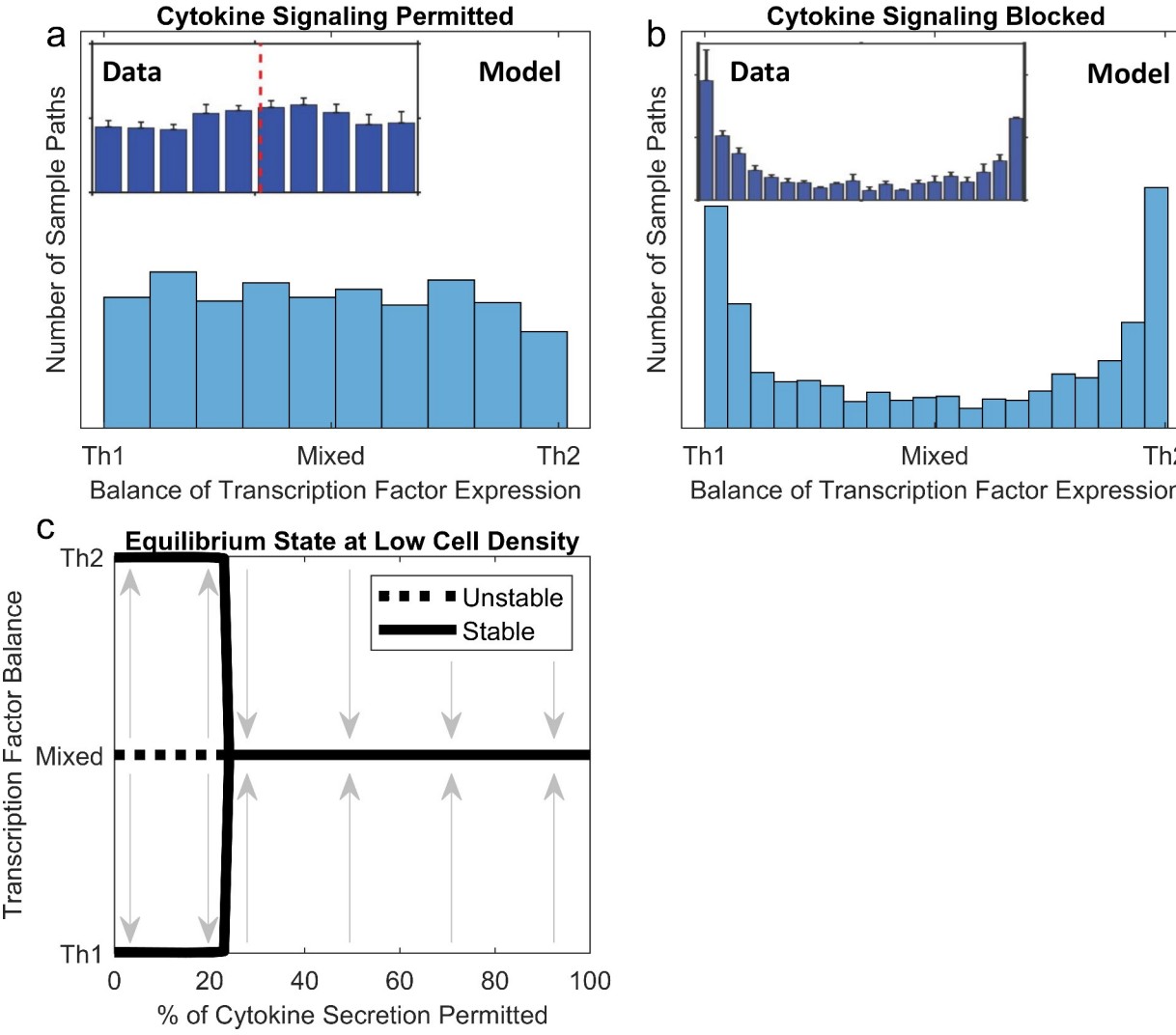

**Fig 2. The model captures the distribution of Th effector types *in vitro* in the presence and absence of cytokine signaling, due to an underlying bifurcation in the dynamical system.** All data are from [34]. Experiments and the model were both run at $2^*10^6$ cells/mL. (a) When cytokines accumulate unhindered, a uniform distribution of Th1, Th2, and mixed effector types is observed, as measured by the balance of T-bet and GATA3 expression, across 1000 sample paths of the SDE system. This closely matches empirical observations (inset). (b) When cytokine accumulation is blocked, a U-shaped distribution of Th1 and Th2 effector types is observed across 1000 sample paths of the SDE system. This also closely matches empirical observations (inset). (c) Analysis of the ODE system shows that mixed effector types are only stable in the presence of cytokine signaling. As cytokine secretion is removed from the model, the mixed effector type becomes unstable and bifurcates into polarized Th1 and Th2 effector types.

would take much longer than feasible Th culture experiments to emerge (S2C–S2F Fig). Thus, our model explains a diversity of experimental results demonstrating that dynamic cytokine signaling drives the balance of mixed Th effector types observed *in vitro*.

## Dynamic cytokine signaling drives polarized Th effector types *in vivo*, via quorum sensing

While cytokine signaling drives mixed effector types at $10^6$ cells/mL, our simple assumptions about cytokine production and removal imply that cytokine dynamics change with cell density (S1 Text). As cell density increases ~1000-fold from *in vitro* (~$10^6$ cells/mL) to *in vivo* (~$10^9$

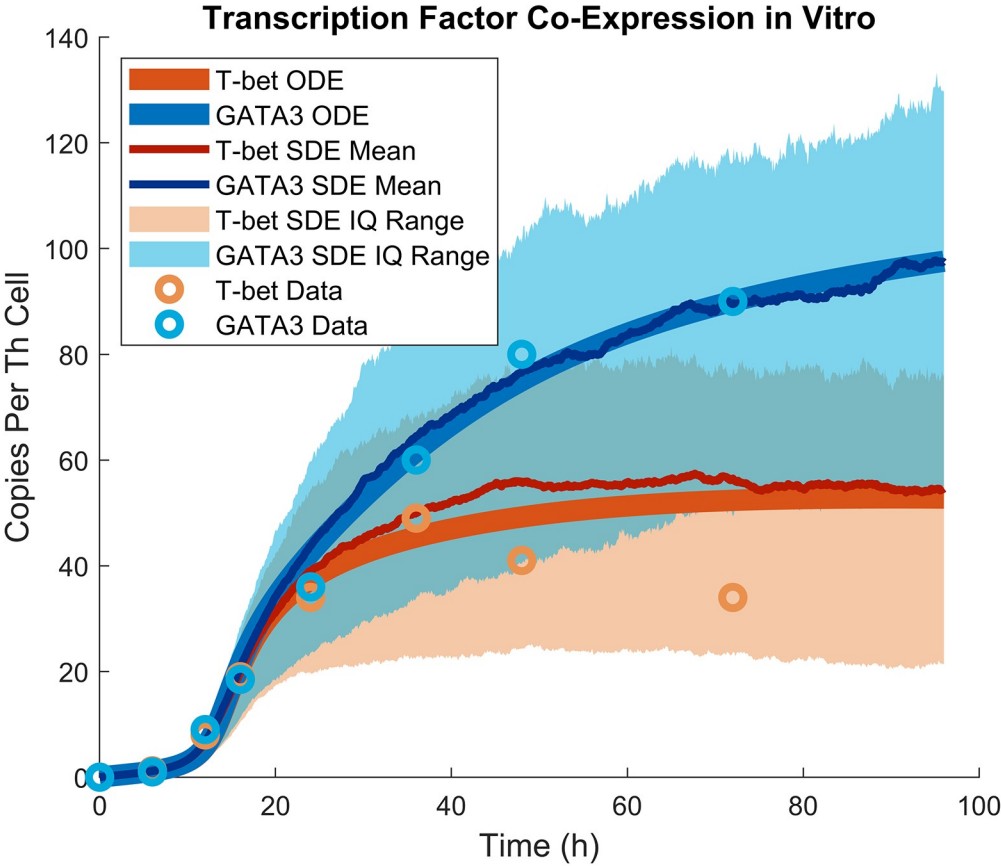

**Fig 3. The model predicts the dynamics of transcription factor expression under *in vitro* conditions with no exogenous effector stimulation.** Experiment and model were both run at $2^* 10^6$ cells/mL. Data are replotted from [34]. SDE mean and interquartile range are drawn from 1000 SDE sample paths.

cells/mL) levels, cytokine concentrations fluctuate faster and accumulate to higher levels (S1 Fig). Faster and stronger cytokine dynamics drives a bifurcation in Th behavior, where mixed effector types become unstable, and the Th group commits fully to either a Th1 or a Th2 effector type (Fig 4A). This contrasts with the previous bifurcation (Fig 2C), in which dynamic extracellular processes were eliminated via cytokine-blocking to decouple the behavior of neighboring Th cells, allowing opposite polarization among individual Th cells. Instead, in this bifurcation (Fig 4A), dynamic extracellular processes are intensified via increased cell density to link the polarization of neighboring Th cells, forcing unified commitments among the Th group. Sensitivity analyses confirm that this result is general and not driven by particular choices of parameter values (S2 Text, S3 and S4 Figs). Because this stark change in group behavior occurs as a function of cell density, this constitutes quorum sensing [6].

Quorum sensing arises naturally from the underlying web of molecular interactions. Among the interactions in our model, some are categorized as "within-scale" because they involve only one spatial scale (i.e. transcription factors directly affecting each other's expression is strictly within cells, and cytokines directly affecting each other's expression is strictly between cells). Other interactions are categorized as "cross-scale" because they involve both spatial scales (i.e. cytokines directly affecting the expression of transcription factors, or vice versa, involves molecules at both the within-cell and between-cell scales). Although some

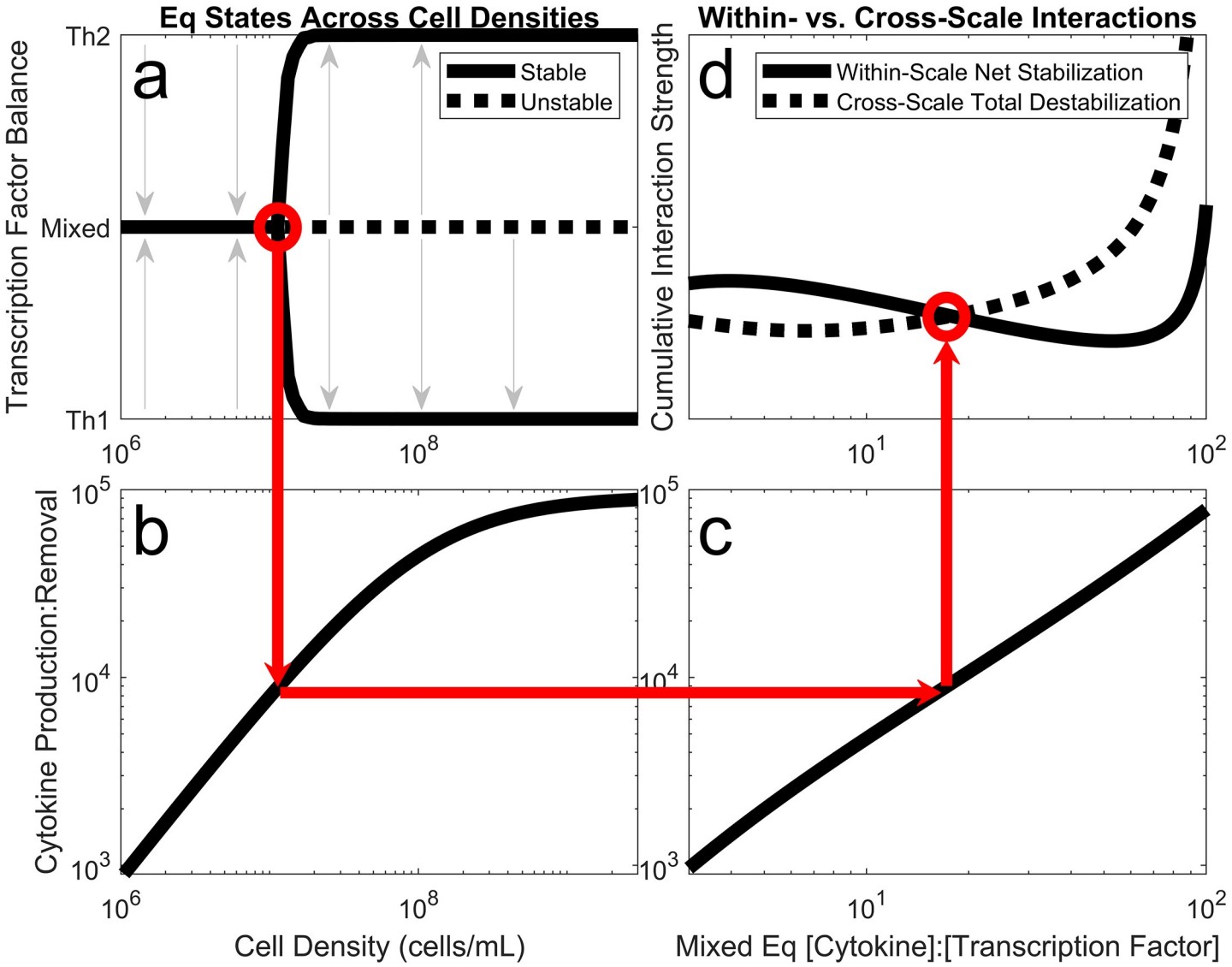

**Fig 4. Th group polarization emerges as cell density increases, due to the changing relative strengths of within-scale vs. cross-scale molecular interactions.** (a) The stable effector balance among a group of Th cells transitions from mixed to polarized as the quorum cell density is surpassed. (b) The quorum cell density corresponds to a particular ratio of cytokine production: removal. (c) This cytokine production: removal ratio controls the ratio of cytokine: transcription factor expression at the mixed effector type equilibrium (y-axis is independent variable and x-axis is dependent variable). (d) The ratio of cytokine: transcription factor expression controls whether the net stabilizing effect of within-scale molecular interactions or the total destabilizing effect of cross-scale molecular interactions is stronger, and therefore whether the mixed equilibrium is stable or unstable.

within-scale interactions favor polarized Th effector types, the net effect of all within-scale interactions together favors mixed effector types. Meanwhile, every cross-scale interaction favors polarized effector types. Therefore, whether the Th group tends toward a mixed or polarized effector type depends on the relative strengths of the within-scale vs. cross-scale molecular interactions (see S4 Text for proof and details). This is consistent with the fact that quorum sensing is inherently a cross-scale phenomenon [1], and it further highlights the importance of dynamic cytokine accumulation, without which there would be no cross-scale interactions.

Because the relative strengths of within- vs. cross-scale molecular interactions depend on cytokine accumulation and therefore on cell density, the emergence of quorum sensing can be explained as follows. The transition in Th group behavior toward unified commitment occurs at the quorum density (Fig 4A). This cell density corresponds to a certain ratio between cytokine production and removal (Fig 4B and S1 Fig). This ratio corresponds to a certain extracellular cytokine concentration at the mixed effector type equilibrium (regardless of whether that equilibrium is stable) (Fig 4C). This extracellular concentration is exactly where the destabilizing cross-scale interactions overpower the stabilizing within-scale interactions (Fig 4D and S4 Text). Among these interactions, only those which are cytokine-mediated change strength with cell density; therefore, it must be the presence of dynamic cytokine signaling in the system which drives quorum sensing. The quorum density is high enough that it is only achieved *in vivo*, leading to self-organized group effector polarization.

## Polarization and frequency of incoming APCs determine speed of Th quorum formation

Having established that biological cell density permits Th quorum sensing, we wondered whether quorum sensing matters during true infections, when APCs are also present. APCs instruct Th effector differentiation via targeted cytokine secretion (see Model Development), which may alter the quorum sensing process. In fact, in our ODE model, even a mixture of Type 1 and Type 2 APCs at very low frequencies sparks the formation of a Th quorum strongly committed to whichever effector type holds a slight majority among the APCs (Fig 5A). However, once a quorum of Th cells is established, even very high frequencies of APCs fully biased toward the opposite effector type cannot overcome the Th quorum to reverse the Th effector choice (Fig 5A). This suggests that APCs play a pivotal role in sparking Th effector choice early in an immune response, but their influence fades as self-organized Th quorum sensing dominates and becomes irreversible.

The transition from APC instruction to Th quorum sensing during the early immune response defines a time window during which APC instruction influences Th effector choice. The duration of this time window is determined by several factors. Most importantly, the more biased incoming APCs are toward a single effector type, the more quickly a Th quorum forms in favor of that type (Fig 5B). Moreover, when APCs are strongly biased, larger numbers of them cause faster quorum formation, although this effect disappears for weakly biased APCs (Fig 5B). This suggests that quorum sensing allows Th cells to translate more confident instruction by APCs into faster effector decisions.

## Cell-to-cell variability in cytokine expression enables Th quora to switch effector types

Because altered APC instruction cannot change the effector choice of a Th quorum, the formation of a Th quorum is irreversible in our ODE model. Irreversibility may be advantageous, as when a growing within-host parasite population manipulates APCs into instructing for the incorrect effector type after some initial time delay. On the other hand, irreversibility may be disadvantageous, as when a secondary coinfection legitimately requires a different effector type in the same lymph node. An optimal Th effector response ought to discern between these two scenarios. Discernment is possible in our model when a Th quorum exhibits cell-to-cell variability in cytokine expression, via stochastic attractor switching.

We demonstrate this discernment by randomly generating time courses of APC instruction, which can range from 100% Th1-biased to 100% Th2-biased (e.g. Fig 6A–6D, gray line). We hypothesize that an optimal Th effector response will ignore sporadic and incomplete

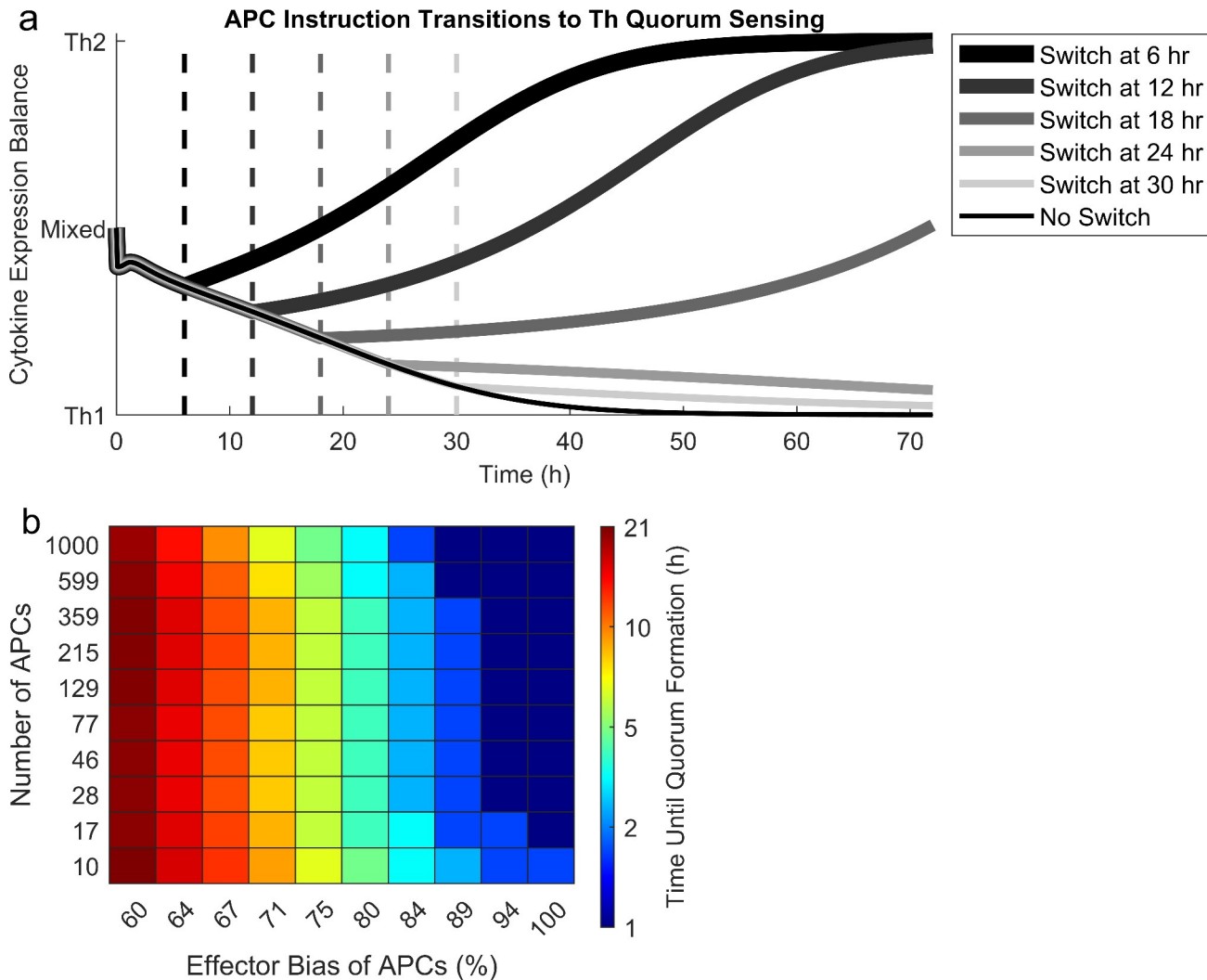

**Fig 5. APCs spark and guide the Th quorum sensing process, but after enough time has passed, they cannot reverse the Th quorum decision.** (a) A representative scenario at biological cell density ($10^9$ cells/mL) in which 10 APCs enter a lymph node, six instructing for Th1 and four instructing for Th2. Even this small Th1 bias among APCs sparks a Th1-committed Th cell quorum. At some time marked by the grayscale dashed lines, the number of APCs increases to 1000, all instructing for Th2. This drastic switch can only reverse the commitment of the Th quorum if it occurs soon after initial arrival of APCs in the lymph node (here ~20 hr). (b) The time required for Th quorum commitment to become irreversible is most strongly controlled by the initial APC effector bias–the higher the percentage of APCs in favor of one effector type, the sooner the resulting Th quorum becomes irreversible. Timing is also impacted by the initial number of APCs–the larger the number of APCs, the sooner the resulting Th quorum becomes irreversible.

changes to APC signaling, which may reflect parasitic manipulation, but will obey sustained and complete changes to APC instruction, which may reflect a legitimate new infection (Fig 6A–6D, black line). In contrast to this hypothesis, the ODE version of our model predicts that a Th quorum will ignore all changes to APC instruction, even sustained and complete changes (Fig 6A, green line). The SDE version of our model, which incorporates cell-to-cell variability in molecular expression, shows that such disadvantageous behavior can be overcome. Low stochastic variability in transcription factor and cytokine expression does not alter the basic result that the Th quorum commitment is irreversible by APCs (Fig 6B). Larger stochastic variability, on the other hand, permits the Th quorum to ignore sporadic and incomplete changes to APC instruction, while obeying sustained and complete changes, albeit with some time lag (Fig 6C). This matches our hypothesis for optimal Th quorum behavior. If stochastic variability is

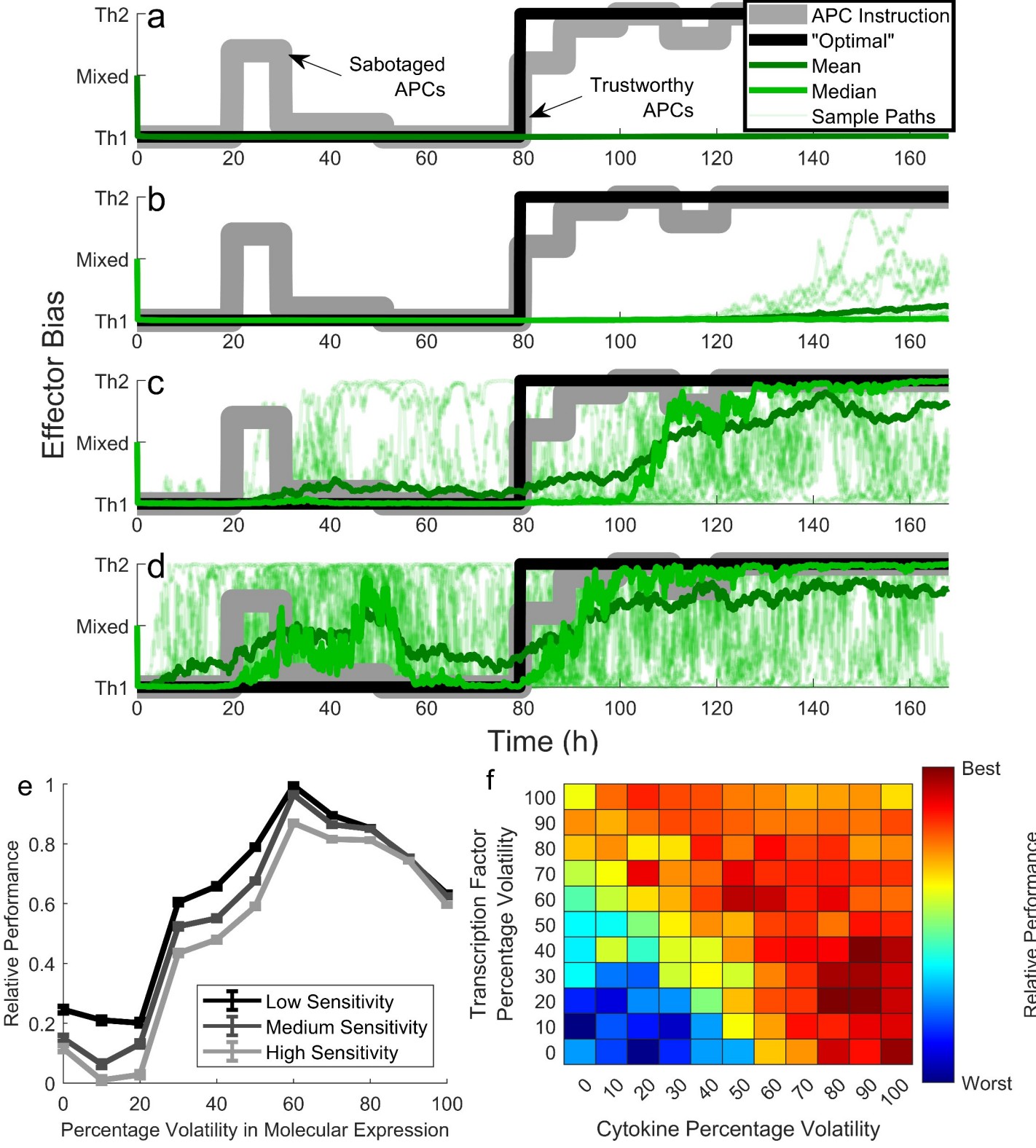

**Fig 6. Cell-to-cell variability in molecular (esp. cytokine) expression allows Th quora to discern when to switch effector types.** All simulations were run at $10^9$ cells/mL. (a) Simulated time-courses of APC effector instruction may include transient and/or sustained changes in instruction (gray line), which Th quora ought to ignore and/or obey, respectively (black line). In the ODE model, Th quora cannot obey even sustained changes to APC instruction. (b) In the SDE model with low levels of variability, Th quora struggle to obey sustained changes to APC instruction. Twenty sample paths along with their mean and median (shades of green) are shown. (c)

Medium levels of variability permit discernment by ignoring transient changes to APC instruction but obeying sustained changes. (d) High levels of variability begin to diminish discernment. (e) Th discernment peaks for intermediate levels of stochasticity in molecular expression, regardless of sensitivity (i.e. how long a change in APC instruction must last before it is considered "sustained"). Percentage volatility = $100^* n_{TF1} = 100^* n_{TF2} = 100^* n_{CY1} = 100^* n_{CY2}$. Relative performance scores how well Th quora tracked the theoretically optimal response, relative to the quorum that did best, across 200 randomly generated time-courses of APC instruction. (f) The analysis shown in (e) was repeated, but where $n_{TF1} = n_{TF2}$ need not equal $n_{CY1} = n_{CY2}$. Data from (e) appear along the diagonal; new data are contained off the diagonal. Th quora perform best when stochasticity in cytokine expression is high, and stochasticity in transcription factor expression is low.

increased further, the Th quorum obeys even transient changes in APC instruction and can even switch effector types at random (Fig 6D), nullifying the benefit of cell-to-cell variability.

This suggests an optimal level of stochastic variability in molecular expression. To test this, we randomly generated 200 time-courses of APC instruction, which may or may not include sporadic and/or sustained changes, and we mapped an optimal Th quorum response to each (as in Fig 6A–6D, gray line and black line). We then measured how well 100 SDE sample paths tracked each APC time-course, across a range of magnitudes of stochastic cell-to-cell variability. Indeed, an intermediate magnitude of stochasticity optimized the Th quorum's ability to track the desired response (Fig 6E). This result holds regardless of assumptions of sensitivity, i.e. how long a change in APC instruction must last before it is considered "sustained" and therefore prudent to obey.

This putatively optimal level of stochasticity in molecular expression raises the question: does stochastic variability in the expression of different types of molecules contribute equally to Th discernment? To answer this question, we repeated the previous analysis but varied the level of stochasticity in transcription factor and cytokine expression independently. This revealed that the performance of the Th quorum is truly optimized when stochasticity in cytokine expression is quite high, while stochasticity in transcription factor expression is quite low (Fig 6F). This suggests that cell-to-cell variability in cytokine expression may be important for effector-type switching in the Th quorum, to discern trustworthy from untrustworthy changes in APC instruction.

## Discussion

Quorum sensing, and other forms of swarming, have repeatedly evolved across various taxa to allow groups of organisms to collectively navigate their environments [1]. Swarming is particularly useful when information is limited [2,3], changing [4,5], or otherwise uncertain [7,38]. All three qualifiers describe the information regarding effector choice that Th cells receive from APCs, which are rare, mutable in effector type, and even subject to sabotage from parasites [e.g. 27,28,29]. Moreover, Th cells possess a well-known mechanism of dynamic signal accumulation–a requirement for quorum sensing [6] and other forms of swarming–in the form of cytokine secretion and consumption. Finally, Th cells are well-suited to swarming in an evolutionary sense. In most swarms, fitness is measured at the individual level (e.g. one fish in a school), such that the benefit of information-sharing to an individual must outweigh the individual cost of helping conspecific competitors, if swarming is to evolve. To the contrary, the evolutionarily relevant fitness of Th cells is measured at the group scale (i.e. the entire host organism) and is unconstrained by costs to individual Th cells [8]. For all these reasons, a system of quorum sensing among Th cells may be logically expected.

Following this expectation, we modeled the quantitative system by which Th cells integrate potentially conflicting and uncertain information from various sources, including each other and APCs, to make effector choices. We label this system quorum sensing because it requires both dynamic signaling across scales and sufficient cell density. At artificially low cell density, as in cell culture, with signaling among Th cells prohibited, individual Th cells polarize toward Th1 or Th2, but not as a unified collective (Fig 2C and Fig 7A). At artificially low cell density

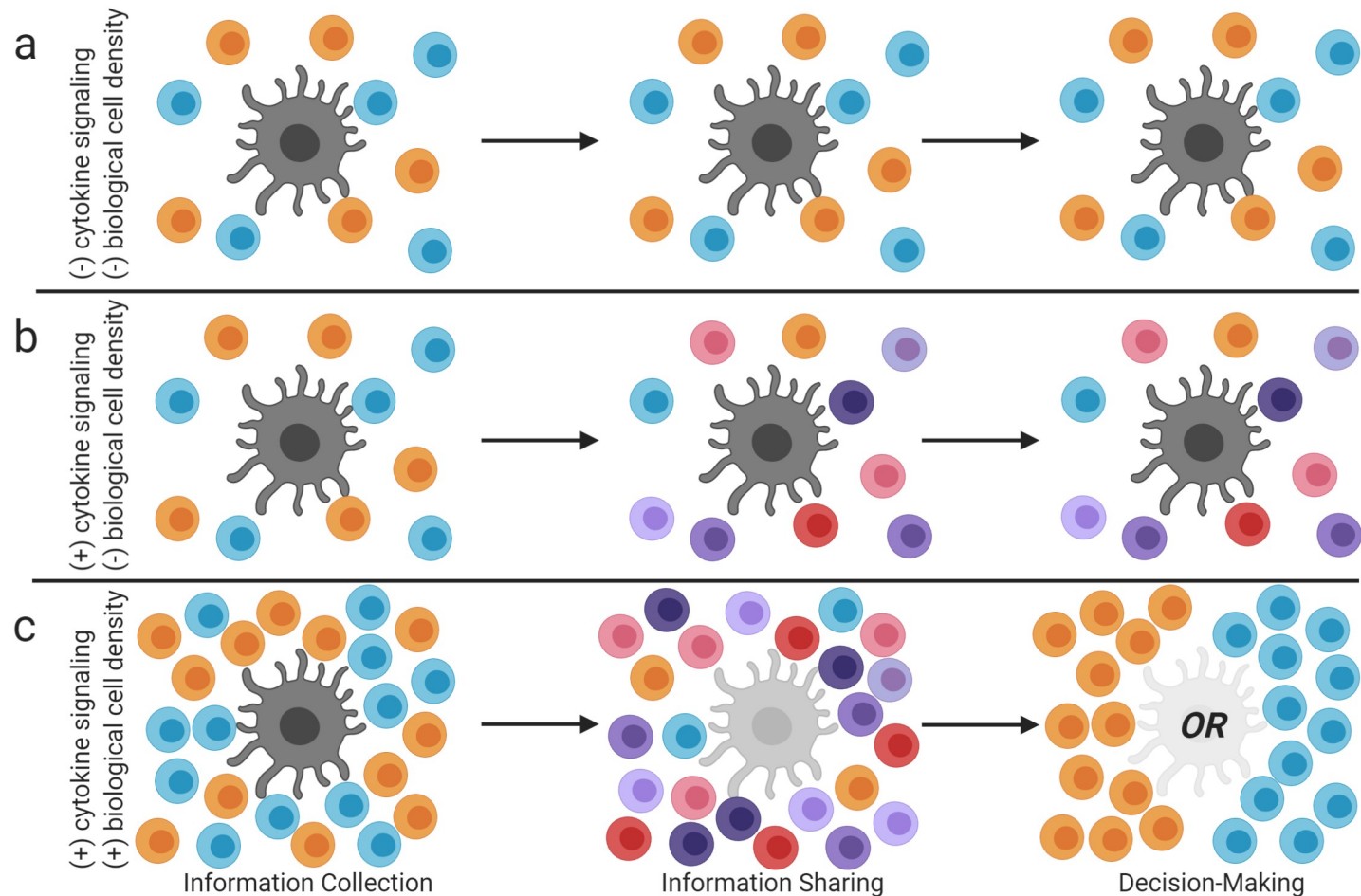

**Fig 7. Cartoon of major conclusions.** Orange circles represent Th1 cells, blue circles represent Th2 cells, and other shades represent Th1-Th2 hybrid cells. The large gray shapes represent APCs, or experimentally provided effector stimulation. (a) At $10^6$ cells/mL with no dynamic cytokine signaling, individual neighboring Th cells adopt oppositely polarized effector types. (b) At $10^6$ cells/mL with dynamic cytokine signaling, oppositely polarized Th cells cause each other to become Th1-Th2 hybrids. (c) At $10^9$ cells/mL with dynamic cytokine signaling, mixed effector types resolve into fully polarized Th1 or Th2 groups, via quorum sensing. Initial polarization by APCs, effector hybrid formation as cytokines dynamically accumulate, and quorum emergence as cytokines accumulate further and APCs are ignored, may define 3 phases of Th effector differentiation *in vivo*.

with signaling among Th cells permitted, cytokines promote stable mixtures of Th1, Th2, and Th1-Th2 hybrid T cells (Fig 2C and Fig 7B). Both conclusions resonate with experimental data (Fig 2A and 2B, and Fig 3). Only at biologically realistic cell density with signaling among Th cells permitted do unified group decisions between Th1 and Th2 effector types emerge (Fig 4A and Fig 7C). This too resonates with various experimental observations [e.g. 30,31,32]. The necessary ingredients for this committed effector choice–dynamic cross-scale signaling among cells and sufficient cell density–define a quorum sensing process.

Because this progression of scenarios also implies a progression of the extracellular cytokine concentration, Th quorum sensing can be understood as a series of phases in time (Fig 7C). Th cells deciding between Th1 and Th2 effector types begin by collecting binary information from APCs. As secreted cytokines accumulate, they invoke molecular feedbacks by which Th cells share information and tune the Th1-Th2 balance of their neighbors. Cytokines continue to dynamically accumulate until they surpass a threshold that is only attainable at biological cell density, precipitating a unified group-level decision between Th1 and Th2 effector types.

This series of phases discounts APC instruction over time, such that the Th quorum decision is eventually irreversible by APC instruction (Fig 5A and 5B). This is consistent with the idea that information gathered by the immune system early in an infection is most trustworthy [54]. Many parasites are capable of manipulating APCs into instructing for the incorrect effector type [28,29]; for example, during infection with *Leishmania* spp., APCs provide appropriate Th1 instruction early in an infection, but later succumb to sporadic manipulation events that alter their effector instructions [27]. Because parasitic manipulation is predicted to influence the evolved structure of immune systems [8,57,58], it may be that quorum sensing is an adaptive parry. If a Th quorum has ceased obeying APC instruction by the time manipulation occurs, then the quality of the host immune response is not compromised, providing robustness in the face of sabotage.

Nonetheless, quorum commitment could be maladaptive when a switch in effector type is truly required. Our model predicts that such switches are possible, when opposing APC instruction is coupled with stochastic variability in molecular expression (Fig 6A–6E). The underlying mechanism, stochastic attractor switching, is observed in other natural systems of collective decision-making, for example by allowing insect swarms to respond to dynamically changing environments [4,5]. Because stochastic attractor switching is a probabilistic phenomenon, the cumulative probability of a transition between states increases with the length of the time window under consideration. Importantly, this allows a Th quorum to discern sustained and legitimate changes in APC instruction from transient and manipulated perturbations to APC instruction. We find that discernment operates best when cytokines, rather than transcription factors, are subject to large cell-to-cell expression variability (Fig 6F). Combined with the observation that exaggerated cell-to-cell variability in cytokine expression is a conserved trait across mammalian species [59], this raises the tantalizing possibility that cytokine expression variability is an adaptive feature of immune signaling [8]. Indeed, signaling variability in other biological swarms, such as house-hunting ants, has already been postulated as an adaptive mechanism to mitigate the speed vs. accuracy tradeoff inherent to decision-making processes with uncertain information [60]. It is possible that natural selection has converged on swarming as a common solution to such problems of uncertainty, both among individual organisms in groups as well as among individual cells within organisms. If the evolution of the mammalian immune system can be understood in this way, then more insights into its organization and functioning may emerge as analogies with other biological swarms are explored further [8,9].

Despite offering a quantitative explanation of Th effector choice that reconciles disparate observations by conceptually unifying collective behavior and immunology, our model does have limitations. For example, our model explains why unified effector choices can emerge *in vivo* but not *in vitro*, and yet unified effector choices are not always observed *in vivo* [e.g. 61]. While our model predicts the long-term equilibrium outcome of Th effector choice, immunity *in vivo* is a non-equilibrium process: cellular birth, migration, and death, parasite replication and death, metabolic inputs and constraints, stochastic events, and a plethora of other factors constantly perturb the immune system. Not every data set will conform to equilibrium predictions, but equilibrium predictions can help explain broad patterns that emerge from the balance of many studies.

Additionally, while the model only addresses Th1 vs. Th2 differentiation, many other effector types exist [18]. In fact, Th differentiation choices between Th17 and iTreg are driven by self-promoting and cross-inhibiting molecular interactions similar to those in this model [62]. Just as this model assumes that a single master transcription factor underlies Th1 and Th2 effector types (T-bet and GATA3, respectively), so too has this "master regulator" assumption been applied to other effector types (e.g. RORγt for Th17, Foxp3 for iTreg) in other

 

mathematical studies to insightfully recover experimentally observed patterns of Th effector differentiation [63,64]. Thus, this model could likely be adapted to represent different or additional Th effector types without changing its basic predictions.

The model also simplifies several details of T cell biology. First, while we have assumed that Th cells exist at roughly $10^9$ cells/mL inside lymph nodes, not all these Th cells actively participate in immunity. In fact, early during infection, only 1 in every $10^5$ or fewer Th cells are activated by any given antigen [65], such that the density of activated Th cells is quite small. However, these activated Th cells proliferate, increasing their numbers by several orders of magnitude [65,66]. Moreover, bystander Th cells which have not been activated by the antigen still participate in effector choice [31]. These processes greatly increase local cell density in the lymph node, likely surpassing the threshold density for quorum sensing. Accounting for Th proliferation might lengthen the information collection and information sharing phases identified by our model, tuning the amount of time until the onset of the decision-making phase (Fig 7C), but it should not preclude quorum sensing altogether. Second, Th cells given consistent effector instruction for long periods of time may undergo epigenetic modifications to commit irreversibly to an effector type, losing plasticity [67]. While our model does not include epigenetic entrenchment, this phenomenon likely requires over a week of stimulation [33] and therefore does not interfere with any of the results we present.

Finally, while our mathematical approach highlights key design principles embedded in the vast complexity of mammalian immunity, direct empirical evidence of quorum sensing in the Th effector choice process remains to be collected. Though technically challenging, experiments that track the effector commitment of individual Th cells over extended time periods given conflicting or fluctuating instructions are needed to test the predictions of this model further. Although unified effector commitment among Th groups may benefit hosts who have coevolved with deceptive parasites, it can also be detrimental. For example, helminth infection can establish an organ-scale commitment to Th2 immunity that prevents vaccines from eliciting proper Th1 memory against deadly intracellular pathogens [31]. Indeed, there is evidence that pre-existing chronic infections consistently diminish vaccine efficacy [68]. Ultimately, we expect that parallel mechanistic and evolutionary understandings of emergent immune phenomena can suggest new ways to manipulate our immune systems, and when it is wise to do so. In turn, successful application of such cross-disciplinary thinking to immunological problems can highlight the power and importance of collectives throughout the natural world.

## Supporting information

**S1 Text. Explanation of model structure, units, and incorporation of cell density.**
(DOCX)

**S2 Text. Sensitivity analyses.**
(DOCX)

**S3 Text. Explanation of the extension to stochastic differential equations.**
(DOCX)

**S4 Text. Mathematical analysis of the model bifurcation.**
(DOCX)

**S1 Table. Parameter values, interpretations, and justifications.**
(DOCX)

**S1 Fig. In terms of concentration in the extracellular environment, [], both cytokine production and cytokine removal rates increase with cell density.** However, production

 

increases faster than removal. Therefore, the ratio of production rate: removal rate increases with cell density.
(TIF)

**S2 Fig. The balance of Th effector types after 1 week in culture can be tuned by the input levels of exogeneous stimulation; however, on longer timescales these effector types converge to a single mixed effector type.** Experiments and model were both run at $2^* 10^6$ cells/mL. (a) T-bet and (b) GATA3 expression, after 1 week in culture, both follow a continuum based on exogeneous stimulation (or blocking via antibodies), just as observed in [33] (c-f) Regardless of the input stimulation and the transient effector balance achieved at 1 week, all conditions converge on a common mixed effector type after approximately 10 weeks.
(TIF)

**S3 Fig. Sensitivity analyses near two parameter points-of-interest, in which parameter values can vary simultaneously near their originally assigned value.** "Asym" can be shown analytically to have no effect on equilibrium position, and therefore marks an effect size that must be insignificant. (a) Near the low-density point-of-interest, the net effect size on equilibrium position of each parameter, controlling for variation in all other parameters, does not exceed 5%. (b) Near the high-density point-of-interest, the net effect size on equilibria position of each parameter, controlling for variation in all other parameters, does not exceed 6%. (c) Near the low-density point-of-interest, the most influential parameters (D and F) exhibit smooth, slight, largely linear, and largely non-interacting effects on the position of the equilibrium. (d) Near the high-density point-of-interest, these parameters still exhibit smooth, slight, largely linear, and largely non-interacting effects on the position of the equilibria.
(TIF)

**S4 Fig. Sensitivity analyses straying from the two parameter points-of-interest, in which parameter values can vary simultaneously up to +/- 90% of their originally assigned value.** (a) Straying from the low-density point-of-interest, new regimes of model behavior (i.e. new numbers of stable equilibria) appear with as little as 20% variation in parameter values, but over half of sampled parameter sets still follow the original model behavior. (b) Straying from the high-density point-of-interest, new regimes of model behavior do not appear even up to 50% variation in parameter values.
(TIF)

## Acknowledgments

We thank S. Forrest, J. Jones, and A. Yates for many discussions that developed the ideas herein. We also thank A. Mayer and C. Wagner for helpful guidance on the mathematics, and A. Yates for insightful comments on the manuscript.

## Author Contributions

**Conceptualization:** Edward C. Schrom, II, Simon A. Levin, Andrea L. Graham.

**Formal analysis:** Edward C. Schrom, II, Simon A. Levin.

**Funding acquisition:** Edward C. Schrom, II.

**Investigation:** Edward C. Schrom, II.

**Methodology:** Edward C. Schrom, II.

**Supervision:** Simon A. Levin, Andrea L. Graham.

**Visualization:** Edward C. Schrom, II.

**Writing – original draft:** Edward C. Schrom, II.

**Writing – review & editing:** Simon A. Levin, Andrea L. Graham.

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
