## [Decision Letter · Decision Letter 0]

10 May 2020

Dear Mr Schrom,

Thank you very much for submitting your manuscript "Quorum sensing via dynamic cytokine signaling comprehensively explains divergent patterns of effector choice among helper T cells" for consideration at PLOS Computational Biology. As with all papers reviewed by the journal, your manuscript was reviewed by members of the editorial board and by several independent reviewers. The reviewers appreciated the attention to an important topic. Based on the reviews, we are likely to accept this manuscript for publication, providing that you modify the manuscript according to the review recommendations.

First, sorry for the time it has taken for the reviews. I agree with both reviewers that the paper is very interesting and with minor modifications will be a very nice publication in PLOS compuational boilogy

Sincerely,

Rustom Antia

Associate Editor

PLOS Computational Biology

Rob De Boer

Deputy Editor

PLOS Computational Biology

[LINK]

First, sorry for the time it has taken for the reviews. I agree with both reviewers that the paper is very interesting and with minor modifications will be a very nice publication in PLOS compuational boilogy

Reviewer's Responses to Questions

**Comments to the Authors:**

Reviewer #1: The review is uploaded as an attachment

Reviewer #2: This is an excellent article that uses a simple model to illustrate how T helper cells make collective choices without central control by using quorum sensing approaches. It is an elegant explanation of differences observed in vivo (heterogenous mixtures of Th1 and Th2) vs polarized choices in vitro driven by a bifurcation in dynamics. The model and analysis explain the role of APC instruction, cytokine signaling and cell density in determining Th cell fate. This work has potential to fundamentally change understanding of effector choice, and perhaps spur other similar investigations into “swarm”-like mechanisms in immune response more generally.

The introduction is concise and compelling and nicely motivates the questions to be addressed in the paper. I suggest including an explanation of quorum sensing which may not be obvious to all readers.

The figures are very clear – starting with Fig 1 that summarizes the dynamics of the model through the last figure that illustrates the what the model means for Th cell choice.

In the pp starting on line 230, I found the cross scale vs within scale terminology confusing. Why not refer to this as within cell vs between cells?

There are several relevant papers to consider citing that describe lymphocytes as a swarm, quorum, and by analogy to ants.

- Switching between individual and collective motility in B lymphocytes is controlled by cell-matrix adhesion and inter-cellular interactions (Theraluz

- Quorum Regulation via Nested Antagonistic Feedback Circuits Mediated by the (Zenke S et al., Immunity, 2020)

- Distributed Adaptive Search in T Cells: Lessons From Ants (Moses et al Fronteirs in Immunology 2019)

**Have all data underlying the figures and results presented in the manuscript been provided?**

Reviewer #1: Yes

Reviewer #2: Yes

PLOS authors have the option to publish the peer review history of their article (what does this mean?). If published, this will include your full peer review and any attached files.

Reviewer #1: No

Reviewer #2: No
---

## [Editor Report · Decision Letter 1]

13 Jun 2020

Dear Mr Schrom,

We are pleased to inform you that your manuscript 'Quorum sensing via dynamic cytokine signaling comprehensively explains divergent patterns of effector choice among helper T cells' has been provisionally accepted for publication in PLOS Computational Biology.

Best regards,

Rustom Antia

Associate Editor

PLOS Computational Biology

Rob De Boer

Deputy Editor

PLOS Computational Biology

---

## [Editor Report · Acceptance letter]

14 Jul 2020

PCOMPBIOL-D-20-00236R1 

Quorum sensing via dynamic cytokine signaling comprehensively explains divergent patterns of effector choice among helper T cells

Dear Dr Schrom,

I am pleased to inform you that your manuscript has been formally accepted for publication in PLOS Computational Biology. Your manuscript is now with our production department and you will be notified of the publication date in due course.

With kind regards,

Sarah Hammond
